# A role for glial fibrillary acidic protein (GFAP)-expressing cells in the regulation of gonadotropin-releasing hormone (GnRH) but not arcuate kisspeptin neuron output in male mice

Charlotte Vanacker[1]*, R Anthony Defazio[1], Charlene M Sykes[1], Suzanne M Moenter[1,2,3,4]*

[1]Departments of Molecular & Integrative Physiology, University of Michigan, Ann Arbor, United States; [2]Internal Medicine, University of Michigan, Ann Arbor, United States; [3]Obstetrics & Gynecology, University of Michigan, Ann Arbor, United States; [4]Reproductive Sciences Program, University of Michigan, Ann Arbor, United States

**Abstract** GnRH neurons are the final central neural output regulating fertility. Kisspeptin neurons in the hypothalamic arcuate nucleus (KNDy neurons) are considered the main regulator of GnRH output. GnRH and KNDy neurons are surrounded by astrocytes, which can modulate neuronal activity and communicate over distances. Prostaglandin E2 (PGE2), synthesized primarily by astrocytes, increases GnRH neuron activity and downstream pituitary release of luteinizing hormone (LH). We hypothesized that glial fibrillary acidic protein (GFAP)-expressing astrocytes play a role in regulating GnRH and/or KNDy neuron activity and LH release. We used adeno-associated viruses to target designer receptors exclusively activated by designer drugs (DREADDs) to GFAP-expressing cells to activate Gq- or Gi-mediated signaling. Activating Gq signaling in the preoptic area, near GnRH neurons, but not in the arcuate, increases LH release in vivo and GnRH firing in vitro via a mechanism in part dependent upon PGE2. These data suggest that astrocytes can activate GnRH/LH release in a manner independent of KNDy neurons.

*For correspondence:
chmhv@umich.edu (CV);
smoenter@umich.edu (SMM)

**Competing interests:** The authors declare that no competing interests exist.

## Introduction

Reproduction is controlled by the hypothalamo-pituitary-gonadal axis. The episodic release of gonadotropin-releasing hormone (GnRH) from neurons in the preoptic area (POA) of the hypothalamus drives the function of this axis (*Belchetz et al., 1978*). GnRH stimulates the anterior pituitary gland to synthesize and secrete two gonadotropins, luteinizing hormone (LH) and follicle-stimulating hormone, which subsequently activate gonadal functions including steroidogenesis. Sex steroids feed back to regulate the GnRH/LH release in both sexes (*Czieselsky et al., 2016*; *Karsch et al., 1987*; *Moenter et al., 1991*; *Hileman et al., 1996*).

How GnRH neurons are activated remains incompletely understood. Over the past several years, kisspeptin neurons located in the arcuate nucleus of the hypothalamus (ARC) have emerged as a leading candidate for initiating GnRH/LH release (*Lehman et al., 2010*; *Navarro et al., 2009*; *Keen et al., 2008*; *Oakley et al., 2009*; *Han et al., 2020*), as well as for integrating steroid negative feedback (*Smith et al., 2005a*; *Smith et al., 2005b*; *Vanacker et al., 2017*; *Moore et al., 2018*). These cells are called KNDy neurons for their coexpression of kisspeptin, neurokinin B, and dynorphin. KNDy neurons in brain slices exhibit changes in spontaneous action potential firing activity that is on the time-scale of episodic LH release and that is steroid-sensitive (*Vanacker et al., 2017*). In

vivo, optogenetic activation of KNDy neurons induces LH release, and increases in intracellular calcium in these cells measured by bulk photometry are associated with spontaneous LH release (*Han et al., 2020*; *Clarkson et al., 2017*). Whether or not KNDy neurons endogenously generate the release of GnRH or are a relay station for a signal generated in other cell types is not known, nor is the list of other cell types that may alter GnRH neuron firing independent of KNDy neurons complete.

In this regard, evidence suggests that astroglia play a role in reproductive function (*Prevot et al., 2005*; *Sandau et al., 2011*; *Clasadonte et al., 2011a*; *Clasadonte and Prevot, 2018*; *Ma et al., 1999*; *Garcia-Segura et al., 2008*). Astrocytes are a major glial type in the central nervous system. GnRH and KNDy neurons are contacted by astrocytes, and astrocytic coverage of neurons in the POA and ARC varies with estrous cycle stage, gonadal status, and seasonal reproduction (*Witkin et al., 1991*; *Baroncini et al., 2007*; *Olmos et al., 1989*; *Cashion et al., 2003*; *Mong and McCarthy, 1999*). Astrocytes can propagate information to adjacent astrocytes (*Nagy and Rash, 2000*; *Hassinger et al., 1996*; *Brancaccio et al., 2017*) making them intriguing candidates for coordinating episodic hormone release over the distances among scattered GnRH neurons. A consensus has yet to emerge regarding vesicle-mediated transmission by astrocytes (*Savtchouk and Volterra, 2018*), but there are data indicating that these cells may release multiple gliotransmitters (*Araque et al., 2001*; *D'Ascenzo et al., 2007*; *Panatier et al., 2011*). Among these is prostaglandin E2 (PGE2), a non-vesicular transmitter that can increase GnRH neuron activity, GnRH release and LH release (*Rage et al., 1997*; *Eskay et al., 1975*; *Harms et al., 1973*).

Our working hypothesis is that local astrocytes modulate GnRH and/or KNDy neuron activity to alter LH release in vivo. To investigate the role of astrocyte signaling in the control of GnRH and KNDy neurons in vitro and LH release in vivo, we took advantage of designer receptors exclusively activated by designer drugs (DREADDs). DREADDs were genetically targeted to astroglia by putting them under the control of the glial fibrillary acidic protein (GFAP) promoter coupled with local adeno-associated virus (AAV) delivery. GFAP is mainly expressed by astrocytes in the central nervous system (*Eng, 1985*), thus this approach allowed temporal and spatial control of G-protein-coupled receptor (GPCR) signaling in astrocytes. Our results indicate that activating signaling in GFAP-expressing cells can induce GnRH neuron activity and LH release in a sex-dependent manner, but does not alter KNDy neuron activity. This suggests that central elements other than KNDy neurons can initiate GnRH release.

## Results

### AAV5 bearing GFAP promoter-driven construct effectively targets GFAP-expressing cells

To examine the response of the reproductive neuroendocrine system to activation of Gq signaling in GFAP-expressing cells, we injected AAV5 bearing GFAP promoter-driven constructs expressing either the mCherry reporter or Gq- or Gi-coupled DREADDs and mCherry. *Table 1* shows the animal models and viruses used in this work and how they are abbreviated in the text; *Table 2* shows the primary antibodies. GFAP is expressed in glial cells throughout the body, including, for example, the pancreas and pituitary (*Regoli et al., 2000*; *Redecker and Morgenroth, 1989*). We thus used stereotaxic injection to limit manipulations to the region of interest. Bilateral viral injections were made in adult mice into either the POA near GnRH soma (*Figure 1A,C*), or ARC near KNDy neurons (*Figure 1B,D*). Successful infection of the area of interest was confirmed by localized expression of

**Table 1.** Mouse strains and viruses used.

|  | Full name | Referred to in text as | Reference | Supplier catalog # | RRID |
|---|---|---|---|---|---|
| Mouse | B6;CBATg(Gnrh1-EGFP)51Sumo/J | GnRH-GFP | *Suter et al., 2000* | JAX 033639 | IMSR_JAX:033639 |
| Mouse | Tg [Tac2-EGFP]381Gsat | Tac2-GFP | *Ruka et al., 2013* | MMRRC; 015495-UCD/STOCK | MMRRC_015495-UCD |
| AAV | AAV5-GFAP-hM3Dq(Gq)-mCherry | AAV-Gq | *Erickson et al., 2021* | Addgene; 50478-AAV5 | Addgene_50478 |
| AAV | AAV5-GFAP-hM4Di(Gi)-mCherry | AAV-Gi | This paper | Addgene; 50479-AAV5 | Addgene_50479 |
| AAV | AAV5-GFAP104-mCherry | AAV-mCherry | This paper | Addgene; 58909-AAV5 | Addgene_58909 |

**Table 2.** Antibodies used.

| Antibody | Species | Dilution/use | Reference | Supplier catalog # | RRID |
|---|---|---|---|---|---|
| Anti-GFAP | Rabbit | 1:10,000 | *Buckmaster et al., 2017* | Agilent DAKO z0334 | RRID:AB_10013382 |
| Anti-S100β | Mouse | 1:500 | *Nishiyama et al., 2002* | Sigma S2532 | RRID:AB_477499 |
| Anti-mCherry | Rat | 1:3000 | *Zhang et al., 2016* | Thermo Fisher Scientific M11217 | RRID:AB_2536611 |
| Anti-NeuN | Rabbit | 1:3000 | *Li et al., 2016* | Abcam ab177487 | RRID:AB_2532109 |
| Anti-GFP | Chicken | 1:1000 | *Kerman et al., 2006* | Abcam ab13970 | RRID:AB_300798 |
| Anti-bovine LHβ 518B7 | | LH assay capture | *Steyn et al., 2013* | Janet Roser, UC Davis | RRID:AB_2665514 |
| AFP240580Rb | Rabbit | LH assay detection | *Steyn et al., 2013* | National Hormone and Peptide Program | RRID:AB_2665533 |
| Goat anti-rabbit HRP | Goat | LH assay enzyme conjugated | *Steyn et al., 2013* | Dako D048701-2 | |

mCherry (*Figure 1A,B*). About 73.3 ± 5.7% of GnRH neurons were surrounded by mCherry in the POA (n=10 mice) and about 90.2 ± 1.7% of kisspeptin neurons were surrounded by mCherry in infected arcuate nuclei (n=13 mice), regardless of virus used. Mice infected unilaterally in the ARC were considered for analysis; it is estimated that 54.7 ± 1.8% of the total population of arcuate kisspeptin neurons (from both nuclei) was surrounded by mCherry in unilateral hit animals (n=3 mice). Dual immunofluorescence for mCherry and either the astroglial cytoplasmic marker S100β (*Figure 1E*) or the neuronal marker NeuN (*Figure 1F*) was performed to characterize the infected population. S100β was used to identify astroglia as GFAP was the driving promoter in the viral constructs used. S100β and GFAP are colocalized in most (<90%) cells examined in hypothalamic tissue (*Figure 1—figure supplement 1*). Data for colocalization of S100β and NeuN with mCherry were similar between ARC and POA and were combined for analysis. Over 90% of mCherry infected cells colocalized with S100β regardless of virus used or region examined (*Figure 1G*). Of note, a few mCherry-positive cells in the ARC region showed morphological characteristics of alpha tanycytes, which are known to express GFAP (*Pellegrino et al., 2018*). A small percent of mCherry infected cells colocalized with NeuN and about 2% of mCherry infected cells did not colocalize with either marker. DREADD expression was thus primarily targeted to astrocytes and compatible with in vitro and in vivo approaches. Because of the small percent of neuronal colocalization, all slices for physiology studies were examined for mCherry infection of neurons based on either morphology or NeuN staining. No expression of mCherry was observed in any recorded or imaged GnRH-green fluorescent protein (GFP) or Tac2-GFP neuron, thus cells from which data were directly obtained were likely not infected. Data from mice (for LH measures) or from those brain slices with mCherry signal observed in cells with neuronal morphology (for electrophysiology) were excluded from statistical analyses. For transparency, these data are shown in different colors (magenta for unidentified neurons, cyan for Tac2-expressing neurons) when individual cell data are presented.

## Gq activation in POA but not ARC GFAP-expressing cells increases LH in vivo in male mice

To examine the response of the reproductive neuroendocrine system to activation of Gq signaling in GFAP-expressing cells, serial blood samples obtained from gonad-intact males were assayed for LH. Injection of vehicle had no effect on LH release in any group (*Figure 2*). When AAV-mCherry control virus was targeted to either the POA or the ARC, CNO injection similarly had no effect on circulating LH levels (*Figure 2A,D*). When AAV-Gq was targeted to the POA, however, CNO induced an abrupt increase in LH by the next 10 min sample (*Figure 2B,C*; two-way, repeated-measures ANOVA AAV-mCherry vs. AAV-Gq $F_{(1,10)}$=59, p<0.0001; time $F_{(8,80)}$=26, p<0.0001; interaction $F_{(8,80)}$=33, p<0.0001). In marked contrast, neither unilateral nor bilateral hits targeting AAV-Gq to the ARC produced a consistent LH response in the absence of overt neuronal infection (*Figure 2E,F*; two-way, repeated-measures ANOVA AAV-mCherry vs. AAV-Gq $F_{(1,13)}$=0.1916, p=0.6688; time $F_{(8,104)}$=1.597, p=0.1345; interaction $F_{(8,104)}$=0.9265, p=0.4980).

We examined LH release in preliminary studies in castrated male mice (n=3/group). CNO had no effect on the mean LH levels (AAV-mCherry, control 4.5±0.7, CNO 3.8±0.5; AAV-Gi, control 4.9±0.3, CNO 4.7±0.5; two-way repeated-measures ANOVA AAV-mCherry vs. AAV-Gi $F_{(1,4)}$=1.4, p=0.2971,

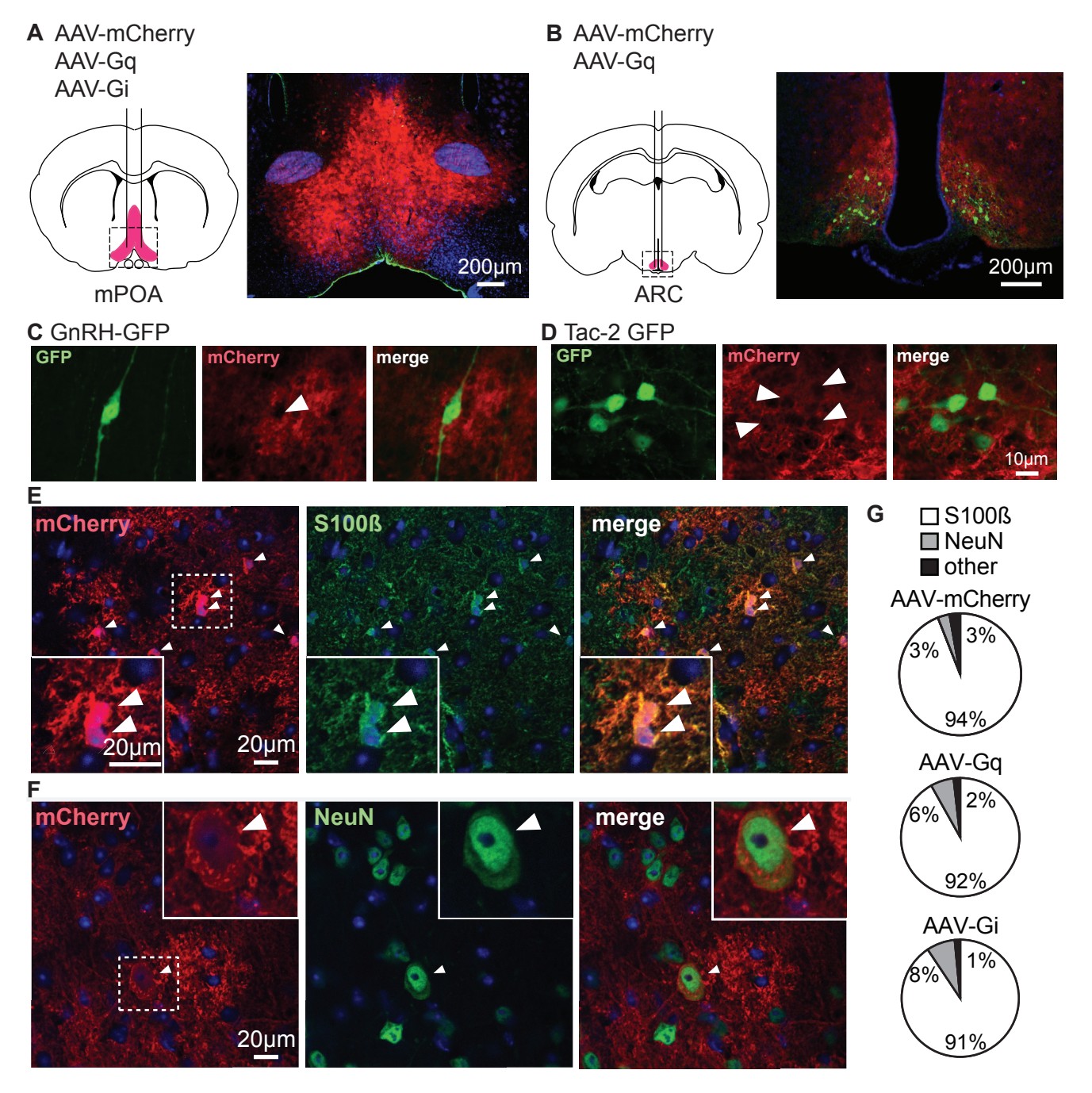

**Figure 1.** Adeno-associated virus 5 (AAV5) bearing glial fibrillary acidic protein (GFAP) promoter-driven constructs effectively target primarily cells with astroglial phenotype and markers in male mice. (**A,B**) Bilateral stereotaxic injection (left) and corresponding infection (right) of the preoptic area (POA) (**A**) and arcuate nucleus of the hypothalamus (ARC) (**B**). (**C,D**) Dual immunofluorescence for green fluorescent protein (GFP) (left) and mCherry (middle) in the POA (**C**) and ARC (**D**) reveals GFP-positive neurons surrounded by infected tissue (merge, right). Arrowheads in middle panels of C, D show gaps in mCherry signal where neurons are located; scale bar is the same for (C) and (D). (**E**) Immunofluorescence for mCherry (left), S100β (middle), and merge (right) showing colocalization of the two signals, white arrowheads identify colocalization between mCherry and S100β staining; dashed box shows area magnified in lower left. (**F**) Immunofluorescence for mCherry (left), NeuN (middle), and merge (right) showing colocalization of the two signals, white arrowheads identify colocalization between mCherry and NeuN staining; dashed box shows area magnified in upper right. (**G**) Quantification of infected cells expressing specific markers for each virus type (n=3 mice per group, 5 fields/mouse).

The online version of this article includes the following source data and figure supplement(s) for figure 1:

**Source data 1.** Data from colocalisation quantification.

*Figure 1 continued on next page*

*Figure 1 continued*

**Figure supplement 1.** Glial fibrillary acidic protein (GFAP) and S100β signals are coexpressed in the majority of cells from male mice (n=3 mice, 5 fields/mouse).

control vs. CNO ($F_{(1,4)}$=6.4, p=0.0631), interaction $F_{(1,4)}$=1.4, p=0.2971) or LH pulse frequency (*Figure 2—figure supplement 1*). Although the p-value for CNO approaches the level accepted for significance, it should be noted that this is most likely attributable to values in the AAV-mCherry, not the AAV-Gi group. Because these preliminary results were negative for LH with the Gi-coupled DREADD, further experiments were restricted to the Gq-coupled DREADD.

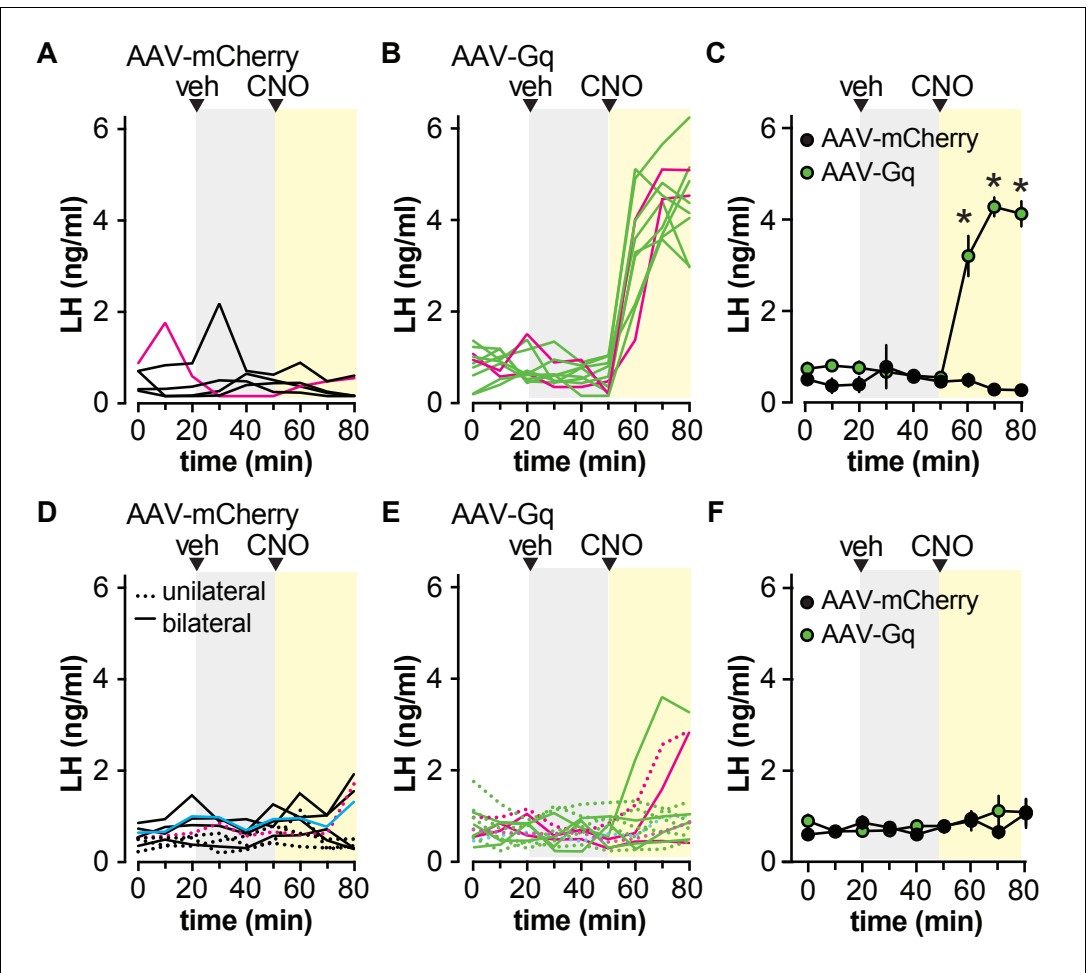

**Figure 2.** Activation of Gq signaling in glial fibrillary acidic protein (GFAP)-expressing cells in the preoptic area (POA) but not the arcuate nucleus of the hypothalamus (ARC) increases circulating luteinizing hormone (LH) in male mice. (**A,B**) LH levels in five mice bilaterally injected in the POA with adeno-associated virus (AAV)-mCherry (**A**) vs. 10 mice injected with AAV-Gq (**B**). Magenta lines in A and B show LH in mice rejected based on infection of unidentified neurons. (**C**) Mean ± SEM LH in mice with no observed neuronal infection *p<0.0001 Bonferroni (n=4–8 mice). (**D,E**) LH levels in individual mice with unilateral (dashed lines) or bilateral (solid lines) ARC hits in nine mice with AAV-mCherry (**D**) vs. 14 mice with AAV-Gq (**E**). Magenta lines in D and E show LH in mice rejected based on infection of unidentified neurons, the cyan line in D indicates a mouse with Tac2 neurons infected. (**F**) Mean ± SEM LH in mice with no observed neuronal infection (n=6–9 mice). Veh, vehicle.

The online version of this article includes the following figure supplement(s) for figure 2:

**Figure supplement 1.** Activating Gi signaling in glial fibrillary acidic protein (GFAP)-expressing cells in the preoptic area (POA) does not affect luteinizing hormone (LH) pulse frequency in male mice in vivo.

## Gq activation in GFAP-expressing cells of the POA increases GnRH neuron firing rate in vitro in male mice

Extracellular recordings of GFP-identified GnRH neurons were made in brain slices from mice infected in the POA with either AAV-mCherry or AAV-Gq. Bath application of CNO (200 nM) had no effect on firing rate of GnRH neurons from AAV-mCherry control mice (*Figure 3A,D*). In contrast, CNO increased firing rate of GnRH neurons within the infected area of AAV-Gq mice as identified by mCherry signal (*Figure 3B,D*, two-way repeated-measures ANOVA mCherry vs. Gq-mCherry F (2,26) = 4.013, p=0.0303). Interestingly, GnRH neurons located outside of the infected region in AAV-Gq mice, easily distinguishable because they were not surrounded by mCherry signal, did not respond to CNO treatment (*Figure 3C*). This suggests that activation of Gq signaling in GFAP-expressing POA cells can increase the firing rate of GnRH neurons but that the propagation of that signal to GnRH neurons in uninfected areas is limited.

To test if activating Gq signaling in GFAP-expressing cells alters firing rate of Tac-2 GFP neurons, the above studies were repeated after targeting injection to the ARC. In this brain region, infection of cells with neuronal morphology within experimental slices was noted. Some of these neurons expressed GFP indicating that they are Tac2-expressing neurons, which are known to activate one another (*Qiu et al., 2016*). While these mCherry-expressing neurons were not recorded, they could influence the response within the slice; data from slices with infected neurons were excluded from statistical analyses but are shown in the individual data plots (*Figure 3E,F*). Neither the type of virus injected (AAV-mCherry vs. AAV-Gq) nor the location of the cell inside or outside the hit affected the firing rate of Tac2-GFP neurons in response to bath application of 200 nM CNO (*Figure 3E–H*, *Table 3*). The firing rate of Tac2-GFP neurons tends to increase over time of recording (*Phumsatitpong et al., 2020*); this was observed as an increase in firing rate during the wash vs. control period.

## CNO-induced increase in GnRH neuron firing rate depends at least in part on activation of PGE2 receptors in male mice

The putative gliotransmitter PGE2 is primarily produced by astrocytes in the hypothalamus and can increase GnRH neuron firing rate (*Ma et al., 1997*; *Clasadonte et al., 2011b*) by acting on EP1 and EP2 receptors expressed by these neurons (*Rage et al., 1997*; *Jasoni et al., 2005*). We hypothesized that the CNO-induced increase in firing in AAV5-Gq mice is PGE2-dependent. We first examined the effects of the stable PGE2 analogue dimethyl-PGE2 (dmPGE2) on firing rate of GnRH and Tac2 neurons. In GnRH-GFP neurons, pretreatment with either DMSO vehicle (0.3%) or a mix of EP1- and EP2-specific antagonists (100 μM SC19220 and 20 μM PF04418948, respectively) had no effect on the firing rate (*Figure 4A–C*). Consistent with previous studies using PGE2 (*Clasadonte et al., 2011b*), dmPGE2 increased GnRH neuron firing rate in cells pretreated with vehicle for the antagonists (*Figure 4A,C*, n=9, p<0.0001 two-way repeated-measures ANOVA/Bonferroni, *Table 4*). Pretreatment with EP1/EP2 receptor antagonists blocked the effect of dmPGE2 on GnRH neuron firing rate (*Figure 4B,C*, n=10, *Table 5*). In contrast to GnRH-GFP neurons, neither pretreatment with methylacetate vehicle nor 200 nM dmPGE2 had an effect on the firing rate of Tac2-GFP neurons (*Figure 4D–F*, n=9 cells, Friedman test F=5.07, p=0.1671).

To test if the CNO-induced increase in firing rate of GnRH neurons in brain slices from mice injected with AAV5-Gq in the POA was dependent upon PGE2 signaling, slices were pretreated with the EP1/2 antagonist mix before exposure to CNO in the continued presence of the antagonists. Pretreatment with antagonists blunted the CNO-induced increase in GnRH neuron firing (*Figure 4F–H*, n=11 cells, p=0.4327, Friedman statistic 2.74, Friedman test). This suggests that activating Gq signaling in GFAP-expressing cells stimulates GnRH neurons at least in part via a PGE2-dependent mechanism.

## Activating Gq did not affect reproductive neuroendocrine parameters in vivo or in vitro in females but there are caveats

To examine if the effects of activating Gq signaling in GFAP-expressing cells is sexually differentiated, females were bilaterally injected in the POA with either AAV-mCherry or AAV-Gq. The survival time post-surgery was longer (3.5–7 vs. 18–22 weeks) because of COVID research shutdown. LH was measured in 10 min samples taken on diestrus. CNO increased LH in

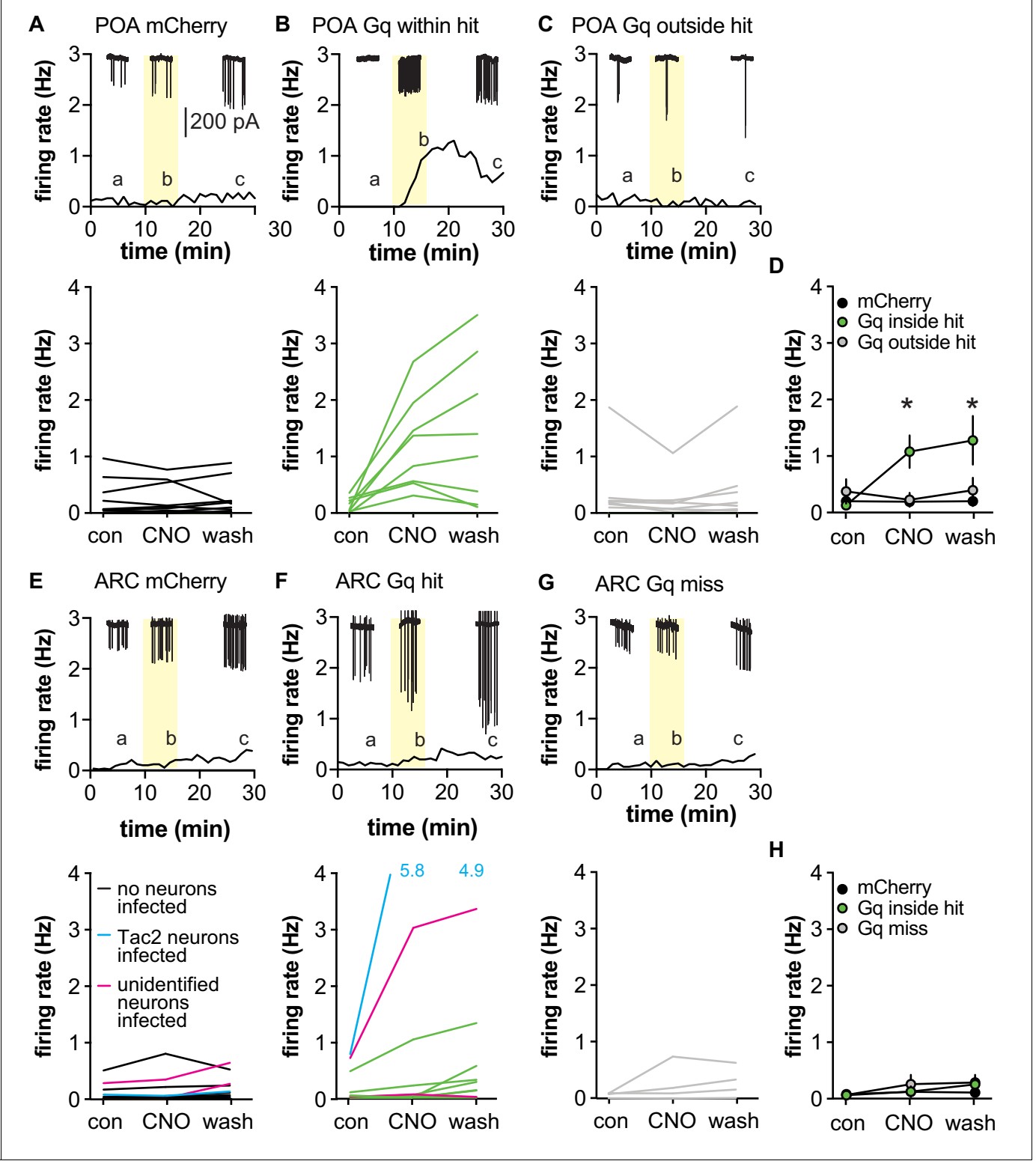

**Figure 3.** Activating Gq signaling in glial fibrillary acidic protein (GFAP)-expressing cells differentially affects gonadotropin-releasing hormone (GnRH) vs. Tac2 neurons in male mice. (A-C) Top, representative example; bottom, firing rate of individual GnRH-green fluorescent protein (GFP) neurons in mice injected in the preoptic area (POA) with adeno-associated virus (AAV)-mCherry (A, n=12 cells from four mice) or AAV-Gq (B, n=9 from six mice, C, n=8 from six mice). Cells in B were within the mCherry-defined hit, cells in C were outside the hit. Representative raw recording data (60 s each, scale bar in A applies to all raw data) from the time indicated by the lower case a,b,c, are shown at the top of each panel in A-C and also E-F. Yellow
*Figure 3 continued on next page*

*Figure 3 continued*

indicated time of CNO treatment. (**D**) Mean ± SEM firing rate during the different periods. *p<0.0001 Bonferroni Gq inside hit vs. Gq outside hit and mCherry (n=8–12). (**E-G**) Top, representative example; bottom, firing rate of individual Tac2-GFP neurons in mice injected in the ARC with AAV-mCherry (**E**, n=12 from five mice) or AAV-Gq (**F**, n=14 from six mice, **G**, n=4 from two mice). Cells in F were within an ARC hit, cells in G were from mice in which the injection missed the ARC. The two magenta lines in E and F indicate cells in slices with unidentified infected neurons; the single cyan line in E and F indicate cells in slices with Tac2 neurons infected. Data from both magenta and cyan cells were excluded from H, mean ± SEM firing rate, *p=0.0033 Bonferroni AAV-Gq control vs. wash (n=4–11).

mice infected with AAV-Gq, but all mice exhibited infected cells with neuronal morphology and these data were not considered further (*Figure 5—figure supplement 1*, virus F(1,9)=12, p=0.0069; time F(9,81)=21, p<0.0001; interaction F(9,81)=25, p<0.0001). In brain slices, CNO had no effect on firing rate of GnRH neurons from female mice infected with either virus (*Figure 5B,C,E*, *Table 5*), unless infected cells with neuronal morphology were observed within the slice (*Figure 5D,E*). These data suggest that there may be a sex difference in the regulation of GnRH neurons by GFAP-expressing cells. An important caveat to point out is that these recordings were made in the afternoon (slices made 2:30–3:30 pm, recordings 3:30–9:30 pm), whereas recordings from males were made in the morning (slices 9:30 am–12 pm, recordings 10:30 am–6 pm).

## Discussion

The central regulation of fertility depends on the secretion of appropriate patterns of GnRH. An emerging dogma in the field is that this is substantially regulated by arcuate kisspeptin, also known as KNDy, neurons. Here, we present evidence that activation of Gq signaling in astrocytes near GnRH neurons in males triggers increased LH release and increased GnRH neuron firing. This is independent of KNDy neurons as similar activation of Gq signaling in astrocytes near these cells fails to alter either LH release or their firing rate. Further, this effect appears to be sexually differentiated as GnRH neurons from diestrous females did not respond. This astrocyte signaling may be a key element in the modulation of GnRH neuron firing and LH release in male mice.

The present findings support and extend previous work suggesting an involvement of astroglia in the regulation of reproductive neuroendocrine function (*Garcia-Segura et al., 2008*; *Sharif et al.,*

**Table 3.** Statistical parameters for effects of CNO on neuronal firing rate (*Figure 3*), p-values <0.05 are in bold.

| GnRH-GFP neurons | F | p | | |
|---|---|---|---|---|
| AAV-mCherry vs. AAV-Gq inside hit vs. AAV-Gq outside hit | (2,26)=4.013 | **0.0303** | | |
| Control vs. CNO vs. wash | (2,52)=7.711 | **0.0012** | | |
| Interaction | (4,52)=8.590 | **<0.0001** | | |
| | *Bonferroni* | *Con vs. CNO* | *Con vs. wash* | *CNO vs. wash* |
| | AAV-mCherry | >0.9999 | >0.9999 | >0.9999 |
| | AAV-Gq in hit | **<0.0001** | **<0.0001** | 0.8009 |
| | AAV-Gq outside hit | >0.9999 | 0>0.9999 | >0.9999 |

| Tac2-GFP neurons | F | p | | |
|---|---|---|---|---|
| AAV-mCherry vs. AAV-Gq inside hit vs. AAV-Gq outside hit | (2,21)=0,2496 | 0.7814 | | |
| Control vs. CNO vs. wash | (2,42)=6.751 | **0.0029** | | |
| Interaction | (4,42)=1.641 | 0.1819 | | |
| | *Bonferroni* | *Con vs. CNO* | *Con vs. wash* | *CNO vs. wash* |
| | AAV-mCherry | >0.9999 | >0.9999 | >0.9999 |
| | AAV-Gq hit | 0.7680 | **0.0033** | 0.0707 |
| | AAV-Gq miss | 0.1342 | 0.0626 | >0.9999 |

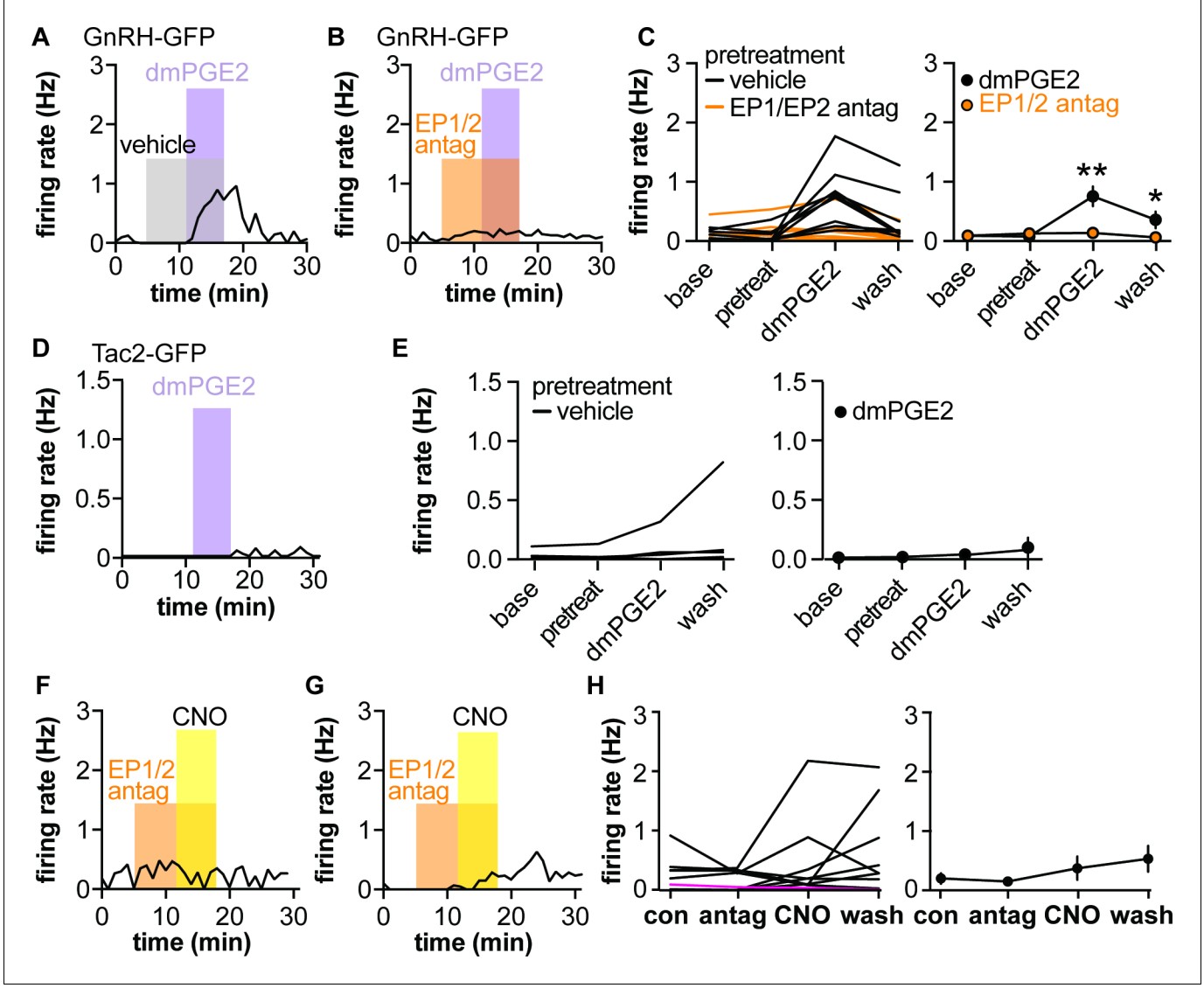

**Figure 4.** Dimethyl-PGE2 (dmPGE2) increases firing rate of gonadotropin-releasing hormone (GnRH)-green fluorescent protein (GFP) but not Tac2-GFP neurons; pretreatment (pretreat) with EP1/EP2 receptor antagonists blunts CNO-induced GnRH neuron firing in male mice. (A,B) Representative examples of GnRH neuron firing rate response to dmPGE2 following vehicle pretreatment (A) or EP1/EP2 receptor antagonists (B). (C) Left, mean firing of individual neurons in each period; right, group mean ± SEM, *p<0.05, **p<0.0001, two-way repeated-measure ANOVA/Bonferroni (vehicle n=9 from five mice, EP1/EP2 antag n=10 from five mice). (D) Representative example of Tac2-GFP neuron response to dmPGE2. Note Y-axis is zoomed in compared to A-C. (E) Left, mean firing of individual neurons in each period; right, group mean ± SEM (n=9). (F,G) Representative examples of GnRH neurons from mice injected with AAV-Gq in the POA in which pretreatment with EP1/EP2 receptor antagonists blocked (F) or reduced response (G) to CNO. (H) Left, mean firing of individual neurons in each period, the single magenta line shows a cell in a slice with AAV-Dq infected neurons (n=12 from seven mice), which was omitted from the group mean ± SEM on the right (n=11 from seven mice).

*2013*; *Ojeda et al., 2010*). Anatomically, a substantial proportion of GnRH neuron somatic membrane is contacted by glia and this contact varies with reproductive state (*Witkin et al., 1991*; *Xiong et al., 1997*). GnRH terminals are intimately connected with specialized tanycytes in the median eminence; this interaction is dependent upon hormonal milieu in females (*Prevot et al., 1999*; *King and Letourneau, 1994*). Functional interactions among GnRH neurons and glia have been postulated to be primarily mediated by PGE2, as blocking central prostaglandin synthesis reduces gonadotropin release (*Ojeda et al., 1975*), whereas injection of PGE2 into the third ventricle or implantation into the POA enhances LH release (*Harms et al., 1973*; *Ojeda et al., 1977*). More recently, PGE2 was shown to increase firing rate of GnRH neurons in both sexes (*Clasadonte et al.,*

**Table 4.** Statistical parameters for effects of dimethyl-PGE2 (dmPGE2), EP1 and EP2 receptor blockers and CNO on gonadotropin-releasing hormone (GnRH)-green fluorescent protein (GFP) neuron firing rate (*Figure 4*), p-values <0.05 are in bold font.

| GnRH-GFP neurons | F | p | | |
|---|---|---|---|---|
| Vehicle vs. EP1/EP2 antagonist pretreatment | F(1,17) | 0.0229 | | |
| Time (baseline vs. pretreatment vs. dmPGE2 vs wash) | F(3,51) | **<0.0001** | | |
| Interaction | (3,51)=11 | <0.0001 | | |
| | *Bonferroni* | *Baseline vs. pretreatment* | *Baseline vs. dmPGE2* | *Baseline vs. wash* |
| | Vehicle | >0.9999 | **<0.0001** | 0.0407 |
| | EP1/EP2 antagonists | >0.9999 | >0.9999 | >0.9999 |

*2011a*). By activating Gq signaling within GFAP-expressing cells directly, the present work extends these findings to the in vivo situation in which the entire hypothalamo-pituitary-gonadal axis can interact, and indicates a role for these cells within the POA in ultimately increasing LH release in males. In other brain regions, when astrocytes were activated with either DREADDs or by agonists of native GPCRs, the time course of increases in intracellular calcium are similar to changes in firing rate in the present study (*Bonder and McCarthy, 2014*; *Kang et al., 2020*; *D'Ascenzo et al., 2007*). This suggests that the GnRH neuron response may be dependent upon the elevation of calcium in astrocytes, and also that the signal conveyed to the GnRH neuron produces a prolonged response consistent with activation of a GPCR within the neurons. This postulate was confirmed as the CNO-induced increase in firing rate response was mimicked by treatment with a stable form of PGE2 and was largely dependent upon activation of EP1 and EP2 receptors.

Astroglia perform several functions within the central nervous system, including release of multiple substances that can serve as gliotransmitters. Astrocytes are also able to communicate over broad areas, propagating elevations in intracellular calcium (*Brancaccio et al., 2017*). This latter observation was key to sparking our interest in how astroglia may function in GnRH release, in particular we postulated that the propagation of signals to distal astrocytes may serve as a mechanism for coordinating activity among the soma of GnRH neurons, which are spread over a wide area from the diagonal band of Broca through the medial basal hypothalamus (*Silverman, 1994*). It was with this postulate in mind that we recorded from GnRH neurons outside the hit region as defined by mCherry fluorescence. These neurons outside the area of infection did not respond to bath application of CNO with an increase in firing rate, however, suggesting that astroglia do not effectively propagate a signal capable of activating distal GnRH neurons in brain slices. In this regard, most GnRH neurons were located inside the hit area, a number substantially above the relatively small number of these cells needed to support LH release (*Livne et al., 1992*; *Herbison et al., 2008*).

Interestingly, the activation of Gq signaling in GFAP-expressing cells within the arcuate nucleus was without effect on either LH release or firing rate of KNDy neurons, indicating that the effects on GnRH neurons were independent of this cell type that appears to play a key role in other aspects of

**Table 5.** Statistical parameters for effects of CNO on gonadotropin-releasing hormone (GnRH)-green fluorescent protein (GFP) neuron firing rate in diestrous females (*Figure 5*), p-values <0.05 are in bold font.

| GnRH-GFP neurons | F | p | | |
|---|---|---|---|---|
| AAV-mCherry vs. AAV-Gq vs. AAV-Gq in neurons | (2,23)=45.50 | **<0.0001** | | |
| Control vs. CNO vs. wash | (2,46)=45.50 | **<0.0001** | | |
| Interaction | (4,46)=30.11 | **<0.0001** | | |
| | *Bonferroni* | *Con vs. CNO* | *Con vs. wash* | *CNO vs. wash* |
| | AAV-mCherry | >0.9999 | >0.9999 | >0.9999 |
| | AAV-Gq | >0.9999 | >0.9999 | >0.9999 |
| | AAV-Gq with infected neurons | **<0.0001** | **<0.0001** | **0.0023** |

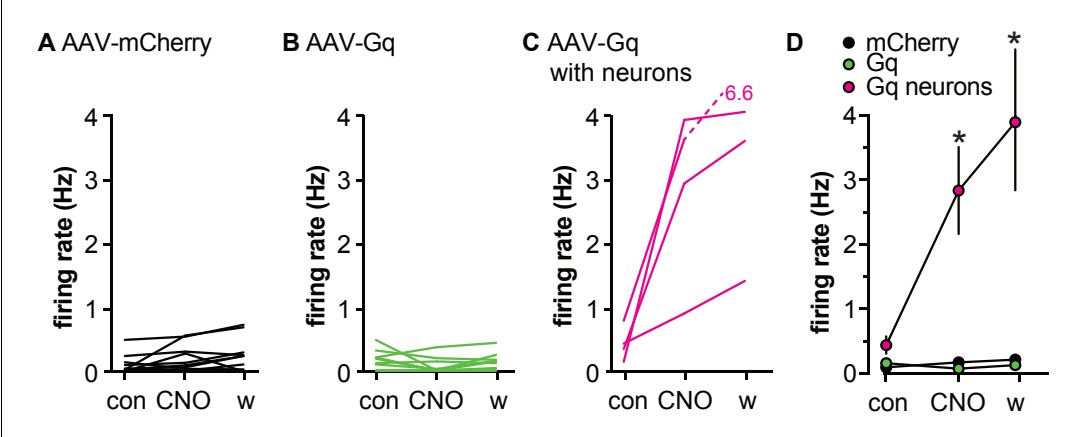

**Figure 5.** Activating Gq signaling in glial fibrillary acidic protein (GFAP)-expressing cells in the preoptic area (POA) does not affect gonadotropin-releasing hormone (GnRH) neuron firing rate in female mice. (**A-C**) Firing rate of individual GnRH-green fluorescent protein (GFP) neurons in brain slices from diestrous females injected in the POA with adeno-associated virus (AAV)-mCherry (A, n=12 from five mice) or AAV-Gq (B, n=10 from five mice, C, n=4 from two mice); slices in C had viral infection of cells with neuronal morphology. Green symbols and lines show data in AAV-Gq infected slices without detected contamination by neuronal infection; magenta symbols and lines show data in slices with Gq-infected neurons. (**D**) Mean ± SEM firing rate during the three periods, *p<0.0001 Bonferroni for cells in slices with Gq-infected neurons vs. the other groups (n=4–12).
The online version of this article includes the following figure supplement(s) for figure 5:

**Figure supplement 1.** Activation of Gq signaling in glial fibrillary acidic protein (GFAP)-expressing cells in the preoptic area (POA) of female mice increases circulating luteinizing hormone (LH) in vivo, but there are caveats associated with infection of cells with neuronal morphology.

GnRH neuron activation. Consistent with a role for PGE2 in the putative astroglia-GnRH communication, treatment of KNDy neurons with PGE2 had no effect upon their firing rate. The lack of effect of astroglia Gq-signaling on KNDy neurons is not likely attributable to a lack of viral coverage as the majority of KNDy neurons were located within the infected area. Interestingly, the area infected by virus includes tissue through which the processes of GnRH neurons pass *en route* to the median eminence. This suggests that this region of the GnRH neuron axon may not be sensitive to mediators released by astroglia. Of interest in this regard, treatment of explants containing the median eminence along with the medial basal hypothalamic region with PGE2-induced GnRH release (*Ojeda et al., 1988*). Our arcuate hits typically did not extend into the median eminence itself. As was the case with GnRH neurons outside the hit, it is possible that there is not sufficient propagation of astroglial signals effective in activating GnRH neuron terminals outside the local region of infection. It is also possible that astrocytes exhibit regional variation in the gliotransmitters they produce. Of note, however, activation of Gq signaling in these cells would be postulated to induce release of whatever substances are downstream of that signaling pathway in the cells and not be limited to prostaglandin synthesis. Together these observations suggest the effects of Gq signaling within astroglia are at least in part specialized to preoptic regions of GnRH neurons for the reproductive neuroendocrine system.

We conducted limited studies of activating Gq signaling in astroglia in diestrous females. CNO treatment failed to alter GnRH neuron firing rate unless there were overtly infected neurons within the slice. There are several possible explanations for these observations. First, cycle stage may affect the response. Second, different gliotransmitters may be released upon astroglia activation in females, perhaps also dependent upon cycle stage. In prior work, over two-thirds of GnRH neurons increased firing rate in response to PGE2 regardless of sex or of cycle stage in females (*Clasadonte et al., 2011a*), indicating responsiveness to this mediator is consistent among groups. Third, it is possible that the relevant astroglia regulating GnRH release are located in different brain regions, for example, more caudal within the anteroventral periventricular region known to be important for induction of the female-specific preovulatory GnRH and LH surge (*Sinchak et al., 2020*).

In contrast to the robust serum LH and GnRH neuron firing response to activation of Gq signaling in astroglia, activation of Gi signaling had no effect on LH in castrated mice. The effects of activating

Gi signaling in GFAP-expressing cells vary with brain region. In the hippocampus, increases in intra-cellular calcium were reported (*Durkee et al., 2019*), whereas in the hypothalamic arcuate region, activation of Gq and Gi signaling has opposing effects on food intake (*Chen et al., 2016*). As in females, interpretation of lack of effect must be limited to the experiments conducted, and it is possible that activation of Gi signaling in astroglia in other brain regions will alter the output of GnRH and LH. Interestingly, there are no reports of receptors acting via Gi-coupled receptors within the POA/hypothalamus.

In the present work, the main mediator of the effects of activating Gq signaling in astroglia appeared to be PGE2. GnRH neurons express both EP1 and EP2 receptors (*Rage et al., 1997*), thus PGE2 could be acting directly upon these cells. There are several things we cannot rule out from our studies. First, while astroglia appear to be the primary source of PGE2 in the hypothalamus, it is possible that other cell types also produce this signal. Second, it is possible that mediators other than PGE2 are produced by astroglia in response to activating Gq signaling. In this regard, in addition to prostaglandins, these cells have been shown to release other mediators including glutamate (*Fellin et al., 2004*; *Slezak et al., 2012*; *Corkrum et al., 2019*) and ATP (*Lalo et al., 2014*; *Guthrie et al., 1999*). Finally, any of these mediators may act via intermediate cell types within the brain slice to bring about effects on GnRH neuron firing. An intriguing question that remains to be answered is what substances serve as the endogenous activators of Gq signaling in POA astroglia. Activation of group 1 metabotropic glutamate receptors induces release of transforming growth factor alpha and neuregulins, which activate ErbB receptors on astroglia. This activation leads to activation of cyclooxygenase 2, a rate limiting step in the synthesis of PGE2 from arachidonic acid. Also of interest, an increase in intracellular calcium induced by activation of Gq signaling via the MrgA1 receptor is associated with the release of PGE2 in astrocytes from organotypic cultures of mouse cerebellum (*Forsberg et al., 2017*). Together with our study, this suggests a link between calcium elevation in astrocytes and PGE2 release that remains to be further delineated. While pure speculation, there are several interesting substances active within the reproductive neuroendocrine system that act via Gq-coupled receptors, including neurokinin B, kisspeptin, and GnRH itself.

In summary, the present work defines that activation of Gq signaling in astrocytes can increase GnRH neuron activity and LH release in a sex-dependent manner. Activation of this pathway may activate this system independent of arcuate kisspeptin neurons or may sculpt the response to that or other critical inputs to GnRH neurons to affect ultimately reproduction.

## Materials and methods

All reagents were purchased from Sigma-Aldrich (St. Louis, MO) unless noted.

### Animals

Mouse strains used for this work are summarized in *Table 1*. Mice expressing enhanced green fluorescent protein (GFP) under the control of *Gnrh* promoter (GnRH-GFP mice, JAX 033639) (*Suter et al., 2000*) or Tac2-GFP BAC transgenic mice (015495-UCD/STOCK Tg [Tac2-EGFP] 381Gsat, Mouse Mutant Regional Resource Center http://www.mmrrc.org/) (*Ruka et al., 2013*) were used to identify GnRH and KNDy neurons for recording, respectively. *Tac2* encodes neurokinin B, which is coexpressed with kisspeptin and dynorphin in KNDy neurons. Tac2-GFP-identified cells in brain slices used for recording also express kisspeptin and/or dynorphin at high percentages, supporting their identity as KNDy neurons (*Ruka et al., 2013*). Mice were held on a 14 hr light/10 hr dark light cycle with lights on at 0300 Eastern Standard Time and had ad libitum access to water and chow (Teklad 2916). Adult gonad-intact males and females were used between 60 and 220 days of age. In females, estrous cycle stage was determined by vaginal lavage and confirmed by uterine mass ($\leq$ 80 mg for diestrus) (*Wagenmaker and Moenter, 2017*). Castration efficacy in males was determined by seminal vesicle mass (intact >250 mg, castrate <150 mg). Animals were distributed randomly among groups and, when possible, experiments were performed blinded. The Institutional Animal Care and Use Committee at the University of Michigan approved all procedures.

### Stereotaxic injections

Viruses used in this work are summarized in *Table 1*. Mice were anesthetized with isoflurane to effect and received 5 mg/kg carprofen before the surgery and 24 hr post-surgery for analgesia. AAV5

carrying a payload encoding the hM3Dq (Addgene 50478) or the hM4Di (Addgene 50479) DREADDs or control AAV (Addgene58909) fused to mCherry under the control of the GFAP promoter were used to introduce DREADDs to regions of interest. AAVs (50–100 nL) were administered by bilateral stereotaxic injection into the POA (AP: −2.27 mm; ML: −0.3 and +0.3 mm; DV: −4.55 mm from frontal vein) or ARC (AP: −1.6 mm; ML: −0.2 and +0.2 mm; DV: −5.95 mm from Bregma). Mice were monitored until fully recovered from anesthesia and surgery sites were examined daily for 10 days. Viral infection was allowed to proceed for 3-6 weeks before experiments.

## Immunohistofluorescence

Mice were deeply anesthetized with isoflurane and transcardially perfused with 0.9% NaCl (10 mL) then 10% neutral-buffered formalin for 15 min (~50 mL). Brains were placed into 10% formalin for 4 hr and transferred into 20% sucrose in 0.1 M PBS for cryoprotection for at least 48 hr and until sectioning. Four series of 30 µm free-floating sections were obtained with a cryostat (Leica CM3050S) in 0.1 M phosphate-buffered saline (PBS) pH=7.4, then transferred to antifreeze solution (30% ethylene glycol, 20% glycerol in PBS) for storage at −20℃. Sections were washed three times in PBS, incubated in blocking solution (PBS containing 0.4% Triton X-100, 2% normal goat serum, Jackson ImmunoResearch) for 1 hr at room temperature, and incubated in primary antibody (*Table 2*) diluted in blocking solution for 48 hr at 4℃. Sections were washed three times in PBS and incubated with Alexa-conjugated secondary antibodies for 1.5 hr at room temperature (Molecular Probes and Jackson ImmunoResearch, 1:500). After three washes with PBS, slices were incubated with 300 nM 4′,6-diaminidino-2-phenylindole dihydrochloride in PBS for 10 min at room temperature. Slices were washed three times in PBS, mounted on Superfrost plus slides (Thermo Fisher Scientific) with Pro-Long Gold antifade reagent (Invitrogen) and coverslipped (VWR International). Primary antibodies and dilutions used are in *Table 2*. Images were collected on a Zeiss AXIO Imager M2 (lower magnification) or on a Nikon A1 confocal microscope (colocalization studies). The number of mCherry only, mCherry/S100β, and mCherry/NeuN coexpressing cells was counted from confocal pictures (3.49 µm optical sectioning, same exposure for each signal, levels adjusted to represent the signal observed by eye, 5 fields/mouse in the infected region; AAV-mCherry and AAV-Gq n=3 mice each POA and ARC, AAV-Gi n=4 mice POA). Brightness was increased 30% in pictures used for figures. In addition to immuno-identification, all 300 µm slices for imaging and electrophysiology were examined for infection of cells with neuronal morphology. Data from slices or mice in which infection of neuronal cells was detected were eliminated from statistical analysis but individual data are shown for transparency in the figures.

## Tail tip blood collection

To examine the effect of DREADD activation on LH levels, mice were handled daily for 2 weeks before CNO (Tocris) administration studies or 5 weeks before sampling LH pulses and habituated to IP injection of 0.9% saline for the last 3-4 days. The tip of the tail was nicked and 6 µL of blood was collected and mixed immediately with 54 µL of assay buffer (PBS, 0.05%Tween and 0.2% BSA). Sampling regimen and animal models were selected based on postulated LH response to activating Gq or Gi signaling within GFAP-expressing cells. To test the effects of activating Gq, postulated to be activating, gonad-intact males were sampled every 10 min for 2 hr, with IP injection of 0.9% saline vehicle at 30 min, and IP injection of 0.3 mg/kg CNO at 60 min. To test the effect of activating Gi, postulated to be inhibitory, males were castrated to elevate LH release; 1 week after castration, these mice were sampled every 6 min for 3 hr, with IP injection of 0.3 mg/kg CNO at 90 min.

## LH assay

Tail blood diluted with assay buffer was kept on ice until the end of sampling, then stored at −20℃ until LH assay by the University of Virginia Ligand Assay and Analysis Core (*Steyn et al., 2013*). The capture monoclonal antibody (anti-bovine LHß subunit, 518B7) is provided by Janet Roser, University of California, Davis. The detection polyclonal antibody (rabbit LH antiserum, AFP240580Rb) is provided by the National Hormone and Peptide Program (NHPP). HRP-conjugated polyclonal antibody (goat anti-rabbit) is purchased from DakoCytomation (Glostrup, Denmark; D048701-2). Mouse LH reference prep (AFP5306A; NHPP) is used as the assay standard. The limit of quantitation (functional

sensitivity) was 0.016 ng/mL, defined as the lowest concentration that demonstrates accuracy within 20% of expected values. Coefficient of variation (%CV) was determined from serial dilutions of a defined sample pool. Intraassay CV was 2.2%; interassay CVs were 7.3% (low QC, 0.13 ng/mL), 5.0% (medium QC, 0.8 ng/mL), and 6.5% (high QC, 2.3 ng/mL).

## Brain slice preparation

All solutions were bubbled with 95% $O_2$/5% $CO_2$ for at least 30 min before exposure to tissue and throughout the experiments. At least 1 week after sampling for LH, brain slices were prepared through the hypothalamus as described (*Chu and Moenter, 2005*). The brain was rapidly removed and placed in ice-cold sucrose saline solution containing (in mM): 250 sucrose, 3.5 KCl, 26 $NaHCO_3$, 10 D-glucose, 1.25 $Na_2HPO_4$, 1.2 $MgSO_4$, and 3.8 $MgCl_2$. Coronal slices (300 µm) were cut with Leica VT1200S (Leica Biosystems, Buffalo Grove, IL). Slices were incubated for 30 min at room temperature (~21–23°C) in 50% sucrose saline and 50% artificial cerebrospinal fluid (ACSF) containing (in mM): 135 NaCl, 3.5 KCl, 26 $NaHCO_3$, 10 D-glucose, 1.25 $Na_2HPO_4$, 1.2 $MgSO_4$, 2.5 $CaCl_2$ (pH 7.4), then transferred to 100% ACSF at room temperature for 0.5–4.5 hr before recording. At the end of the recording, slices of interest were fixed with 10% formalin for 40 min, washed three times in PBS, and were processed for immunofluorescence to study the spread of the infected area or examined directly for mCherry signal in cells with neuronal morphology.

## Electrophysiological recordings

Targeted single-unit extracellular recordings were used to minimize impact on the cell's intrinsic properties (*Nunemaker et al., 2003*; *Alcami et al., 2012*). Recording pipettes (2–4 MΩ) were pulled from borosilicate glass (Schott no. 8250; World Precision Instruments, Sarasota, FL) with a Sutter P-97 puller (Sutter Instrument, Novato, CA). Pipettes were filled with HEPES-buffered solution containing (in mM): 150 NaCl, 10 HEPES, 10 D-glucose, 2.5 $CaCl_2$, 1.3 $MgCl_2$, and 3.5 KCl, and low-resistance (6–15 MΩ) seals were formed between the pipette and neuron. Recordings were made in voltage clamp with a 0 mV pipette holding potential, data acquired at 10 kHz, and filtered at 5 kHz using a one amplifier of an EPC10 dual patch clamp amplifier controlled with PatchMaster software (HEKA Elektronic, Lambrecht, Germany).

Slices were transferred to a recording chamber with constant perfusion of carboxygenated ACSF at 29–32°C at a rate of approximately 3 mL/min. GFP-positive cells were targeted for recording in the POA (GnRH neurons) or ARC (KNDy neurons). Cells that were surrounded by mCherry signal were considered within the infected area (hit) and those with no detectable peripheral mCherry signal within 150 µm were considered outside the infected area (miss). Recordings consisted of a 5–10 min stabilization period, a 5 min control period, bath application of treatment for 6 min, followed by a wash. Mean firing rate was calculated for the last 3 min baseline period, for min 5–7 of treatments, and min 8–10 after treatment (wash). At the end of each recording, cells that were inactive throughout the recording were treated with 20 mM potassium in ACSF; cells that exhibited action currents in response were verified to be alive and recordable, and all data were used. Cells that did not respond to elevated potassium were excluded. No more than three cells from the same animal were included and at least four animals were studied per group for electrophysiology studies.

To test the hypothesis PGE2 affects GnRH and KNDy neuron firing, we used 16,16-dimethyl PGE2 (dmPGE2, 200 nM), a stable form of PGE2 (Cayman Chemical 14750) in 0.00076% methyl acetate vehicle. Antagonists for prostaglandin receptor EP1 (SC-19220, 100 µM, Tocris) and EP2 (PF-04418948, 20 µM, Tocris) were diluted into fresh bubbled ACSF (final DMSO concentration 0.3%) and bath applied before CNO treatment. The recording paradigm was a 6 min baseline, 6 min pretreatment with methyl acetate vehicle containing either DMSO vehicle or EP1 and EP2 antagonists, a 6 min treatment that added 200 nM dmPGE2, followed by wash in ACSF. Mean firing rate was calculated for the last 3 min baseline and pretreatment periods, min 5–7 following addition of dmPGE2 for treatment, and min 8–10 after wash began.

## Analysis and statistics

Data were analyzed using code written in IgorPro 8 (Wavemetrics). The code is available on GitLab (*DeFazio, 2021* copy archived at swh:1:rev:2d400d05d3443defb99bf8c8a9b3c2c846a8b1be). Action currents (events) were detected and confirmed by eye. Statistical analyses were conducted with

GraphPad Prism 9. Distribution normality was checked with Shapiro-Wilk; most experiments included one or more groups that were not normally distributed. One-way repeated-measures tests were conducted with the Friedman test. For two-way repeated-measures ANOVA, the Bonferroni post hoc is considered sufficiently rigorous for non-normally distributed data (*Underwood, 1997*). Statistical tests for each study are specified in the Results section. $p < 0.05$ was considered significant and exact p-values are reported.

## Acknowledgements

We thank Elizabeth Wagenmaker and Laura Burger for expert technical assistance.

## Additional information

### Funding

| Funder | Grant reference number | Author |
| --- | --- | --- |
| Eunice Kennedy Shriver National Institute of Child Health and Human Development | R37 HD034860 | Suzanne M Moenter |

The funders had no role in study design, data collection and interpretation, or the decision to submit the work for publication.

### Author contributions

Charlotte Vanacker, Conceptualization, Data curation, Formal analysis, Methodology, Writing - original draft, Project administration, Writing - review and editing; R Anthony Defazio, Software, Methodology, Writing - review and editing; Charlene M Sykes, Data curation, Formal analysis, Writing - review and editing; Suzanne M Moenter, Conceptualization, Resources, Supervision, Funding acquisition, Validation, Investigation, Writing - original draft, Project administration, Writing - review and editing

### Author ORCIDs

Charlotte Vanacker  https://orcid.org/0000-0001-5289-9298
R Anthony Defazio  https://orcid.org/0000-0001-7302-7528
Suzanne M Moenter  https://orcid.org/0000-0001-9812-0497

### Ethics

Animal experimentation: This study was performed in strict accordance with the recommendations in the Guide for the Care and Use of Laboratory Animals of the National Institutes of Health. All of the animals were handled according to approved institutional animal care and use committee (IACUC) protocols of the University of Michigan. The Institutional Animal Care AND Use Committee at the University of Michigan approved all procedures (PRO00006816 PRO00008797).

### Decision letter and Author response

Decision letter https://doi.org/10.7554/eLife.68205.sa1
Author response https://doi.org/10.7554/eLife.68205.sa2

## Additional files

### Supplementary files
- Source data 1. Firing data for all recorded cells.

- Transparent reporting form

## Data availability

Analysis code for calcium signals and event detection have been deposited to GitLab and are accessible at https://gitlab.com/um-mip/coding-project (copy archived at https://archive.softwareheritage.org/swh:1:rev:b9d40510b04e23174dc618b8b31953294ffb3050).

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
