## [Decision Letter]

**Acceptance summary:**

Your study will be of interest to neuroendocrinologists and more broadly to neuroscientists since it addresses the contribution of astrocytes to the activity of a neuronal system orchestrating the reproductive function. You used an innovative techniques to target astrocytes and provide results that complement and extend previous work on the role of astrocytes in the control of the GnRH system.

**Decision letter after peer review:**

Thank you for submitting your article "A role for glial fibrillary acidic protein (GFAP)-expressing cells in the regulation of gonadotropin-releasing hormone (GnRH) but not arcuate kisspeptin neuron output" for consideration by eLife. Your article has been reviewed by 2 peer reviewers, and the evaluation has been overseen by a Reviewing Editor and Catherine Dulac as the Senior Editor. The reviewers have opted to remain anonymous.

Essential revisions:

The results from the experiment examining AAV-Gi effect on LH could be shown in a supplementary figure.

The values of LH dosage in female mice should be provided.

Figure 4B: why is the firing rate equal to 0 before application of CNO while a low frequency firing rate is visible in AAV-mCherry and AAV-Gq outside hits?

To address possible sex differences, in addition to recording GnRH neurons in the same conditions as in males, analysis of Tac2 neurons should also be included in order to extend the impact of the present study.

For better readability, the number of animals used in each experiment should be clearly specified (e.g in the figure legends). Moreover, in neuronal firing rate analyses (data shown in figures 4, 5 and 6), the details of how many cells were recorded from how many animals should be indicated.

For better readability, the sex of animals used in each experiment should be indicated in the main text and figure legends.

Table 2: the column "reference" is empty. Please complete.

Page 9, first paragraph: a reference to Figure 5 is missing.

Please correct the following typos:

- Page 3 line 48: « this approaches ».

- Page 4 line 59: remove the parenthesis.

- page 4 line 68: "cells examined hypothalamic tissue".

- page 5 line 78: "with mCherry signal was observed".

- page 5 line 88: "together with the either".

- page 8 line 168: "AAV-Dq".

What receptor/mechanism is required in astrocytes to produce and release PGE2? Have the authors considered a role for MrgA1 receptors in astrocytes to be a Gq receptor mechanism that can be responsible for the release PGE2 and subsequent alteration of neuronal activity? Possibly recent work from Forsberg and colleagues may be relevant (PMID: 28976306).

Curious about the location of the injections and the impact on the reported observations. For example, in Figure 3 the authors observed an increase in LH in one mouse that was received AAV-Gq bilaterally into the ARC and was verified to be absent of infection in unidentified neurons. Similarly, a single ARC Tac2 neurons appears to have increased firing rate in response to CNO in Figure 4F (in a slice absent of unidentified or Tac2 infected neurons). What may explain this response while the remaining animals did not exhibit this increase. Did the authors observe any rostral to caudal or medial to lateral differences with regard to the injections in either the ARC or the POA? In addition to injection location, did the authors observe any differences in penetrance of virally transduced neurons?

PGE2 in the POA is associated with both thermogenic and tachycardic conditions (PMID: 2245331; PMID: 12648744; PMID: 16406311; PMID: 1967907). Do the authors suspect release of PGE2 from astrocytes in the POA regulates this activity?

Figure 2B and C. the authors should define the color coding. What is the significance of the black, cyan, and magenta traces in each panel? Are these different ROIs?

Figures 3, 4, 5; the overlap of the magenta and cyan lines with green lines is confusing. Possibly the authors could modify these figures based on the format of figure 6 which shows separate graphs for infected neurons.

The number of animals used for the current study is surprisingly small (n=1 or 2; table 3).

What age animals were used?

[Editors' note: further revisions were suggested prior to acceptance, as described below.]

Thank you for resubmitting your work entitled "A role for glial fibrillary acidic protein (GFAP)-expressing cells in the regulation of gonadotropin-releasing hormone (GnRH) but not arcuate kisspeptin neuron output in male mice" for further consideration by eLife. Your revised article has been evaluated by Catherine Dulac (Senior Editor) and a Reviewing Editor.

The manuscript has been improved but there are some remaining issues that need to be addressed, as outlined below:

The calcium imaging analyses performed on 1 to 2 animals per group should be deleted or additional cases should be added.

---

## [Author Response]

Essential revisions:The results from the experiment examining AAV-Gi effect on LH could be shown in a supplementary figure.

These data are now provided in Figure 3-figure supplement 1.

The values of LH dosage in female mice should be provided.

These data are now provided in Figure 6-figure supplement 1.

Figure 4B: why is the firing rate equal to 0 before application of CNO while a low frequency firing rate is visible in AAV-mCherry and AAV-Gq outside hits?

We chose our cells for illustration based on a representative response to CNO, which the cell in Figure 4B had. There were multiple cells that did not fire during the control period; this can be appreciated by the individual data in the lower panels of Figures 4A and B. While there were no cells in the “outside hit” group (Figure 4C) that were completely quiescent before treatment, these cells fell well within the typical GnRH neuron firing behavior and were not different from the other groups.

To address possible sex differences, in addition to recording GnRH neurons in the same conditions as in males, analysis of Tac2 neurons should also be included in order to extend the impact of the present study.

We agree with the reviewers that this is a logical extension of the present work, as is a re-examination of the effects on GnRH neurons in females. But we view this as a new project as it is a considerable amount of additional work (likely at least a year). This area of investigation has transitioned from the first author (a former postdoc in the lab) to a new graduate student who is now developing his proposal. We believe there is sufficient new information for the field in the work presented. We agree that the female work is limited by caveats, but we want to be transparent in our findings and provide the information, such as it is with caveats pointed out, to others who may be working on similar projects to assist with their experimental development given the amount of time it is likely to take before an update is ready for publication. Similarly, it is our lab goal to have graduate students design their own studies (with mentoring) and to simply train the new student and have them serve as ‘hands’ isn’t in keeping with this goal.

For better readability, the number of animals used in each experiment should be clearly specified (e.g. in the figure legends). Moreover, in neuronal firing rate analyses (data shown in figures 4, 5 and 6), the details of how many cells were recorded from how many animals should be indicated.

The number of animals and number of cells per animal have been added to the legend for each figure.

For better readability, the sex of animals used in each experiment should be indicated in the main text and figure legends.

Sex of the animals has been added to figure legends and clarified in the main text.

Table 2: the column "reference" is empty. Please complete.

Thank you for pointing this out, references have now been included.

Page 9, first paragraph: a reference to Figure 5 is missing.

Thank you, this has been added.

Please correct the following typos:- Page 3 line 48: « this approaches ».- Page 4 line 59: remove the parenthesis.- page 4 line 68: "cells examined hypothalamic tissue".- page 5 line 78: "with mCherry signal was observed".- page 5 line 88: "together with the either".- page 8 line 168: "AAV-Dq".

Corrected.

What receptor/mechanism is required in astrocytes to produce and release PGE2? Have the authors considered a role for MrgA1 receptors in astrocytes to be a Gq receptor mechanism that can be responsible for the release PGE2 and subsequent alteration of neuronal activity? Possibly recent work from Forsberg and colleagues may be relevant (PMID: 28976306).

As the reviewers point out, the endogenous receptor required in astrocytes is an important point for future consideration. We had discussed this (lines 305-309) and have expanded our discussion to include the work on MrgA1 and the Forsberg paper.

Curious about the location of the injections and the impact on the reported observations. For example, in Figure 3 the authors observed an increase in LH in one mouse that was received AAV-Gq bilaterally into the ARC and was verified to be absent of infection in unidentified neurons. Similarly, a single ARC Tac2 neurons appears to have increased firing rate in response to CNO in Figure 4F (in a slice absent of unidentified or Tac2 infected neurons). What may explain this response while the remaining animals did not exhibit this increase. Did the authors observe any rostral to caudal or medial to lateral differences with regard to the injections in either the ARC or the POA? In addition to injection location, did the authors observe any differences in penetrance of virally transduced neurons?

These are very good questions that we considered in shaping our manuscript. With regard to injection location, the POA was more consistent than the arcuate likely because it provides a considerably larger target. We evaluated rostral caudal infection in the ARC but this did not account for variability. Similarly, we sometimes observed stronger infection penetrance on one side than the other, but again could discern no difference attributable to this. As for the ARC-infected animal that responded but did not have detectable infection of cells with neuronal morphology, we looked for anything out of the ordinary but found nothing technical that could explain this result. One possibility is that a small percent of ARC kisspeptin neurons do respond to glial factors. Another is that there was some terminal infection of GnRH fibers (not observed but perhaps not detectable). Finally, it is possible that this animal did have infection of cells with neuronal morphology but that this could not be detected from the mCherry signal.

PGE2 in the POA is associated with both thermogenic and tachycardic conditions (PMID: 2245331; PMID: 12648744; PMID: 16406311; PMID: 1967907). Do the authors suspect release of PGE2 from astrocytes in the POA regulates this activity?

We did not make these measurements in animals injected with viruses to target DREADD expression. Very early studies for this project were done using a GFAP-cre mouse crossed with floxed Gq-DREADD. These *in vivo* studies were dropped promptly because of adverse systemic effects, including reduced body temperature noticeable to the touch and elevated blood glucose. We noticed no behavioral changes upon injection of CNO in virus-injected mice and did not monitor them further.

Figure 2B and C. the authors should define the color coding. What is the significance of the black, cyan, and magenta traces in each panel? Are these different ROIs?

These are indeed ROIs and we have added this to the legend.

Figures 3, 4, 5; the overlap of the magenta and cyan lines with green lines is confusing. Possibly the authors could modify these figures based on the format of figure 6 which shows separate graphs for infected neurons.

We tried this suggestion but decided to keep the original design for different reasons on each figure. For Figure 3, there are more frequent data points, thus ‘squishing’ the data in the x axis direction to permit separation reduces the size of the relevant data more than is ideal. For Figure 4, there are only three cells each in the bottom panel of parts E and F and these can be clearly seen; we have added the number of these cells to the legend so the reader can be confident of having identified them. We took the same approach with the single cell in 5H.

The number of animals used for the current study is surprisingly small (n=1 or 2; table 3).

Table 3 is for calcium imaging studies. The n here is low because this was a technical control to confirm previously published work could be repeated in our lab. Number of animals for the other studies is a minimum of four per group (line 429-430).

What age animals were used?

Ages were provided on line 327 in the animals section of the Materials and Methods.

[Editors' note: further revisions were suggested prior to acceptance, as described below.]

The manuscript has been improved but there are some remaining issues that need to be addressed, as outlined below:The calcium imaging analyses performed on 1 to 2 animals per group should be deleted or additional cases should be added.

These data and all references to them have been deleted as requested.